

# Potential lineage transmission within the active microbiota of the eggs and the nauplii of the shrimp *Litopenaeus stylirostris*: possible influence of the rearing water and more

Carolane Giraud[1,2], Nolwenn Callac[1], Maxime Beauvais[1,3], Jean-René Mailliez[1], Dominique Ansquer[1], Nazha Selmaoui-Folcher[2], Dominique Pham[1], Nelly Wabete[1] and Viviane Boulo[1,4]

[1] Ifremer, IRD, Université de la Nouvelle-Calédonie, Université de La Réunion, CNRS, UMR 9220 ENTROPIE, Noumea, New Caledonia
[2] University of New Caledonia, Institut des Sciences Exactes et Appliquées (ISEA), Noumea, New Caledonia
[3] Sorbonne Université, UMR 7261, Laboratoire d'Océanographie Microbienne, Observatoire Océanologique de Banyuls-sur-Mer, CNRS, Banyuls-sur-Mer, France
[4] IHPE, Université de Montpellier, CNRS, Ifremer, Université de Perpignan via Domitia, Montpellier, France

## ABSTRACT

**Background.** Microbial communities associated with animals are known to be key elements in the development of their hosts. In marine environments, these communities are largely under the influence of the surrounding water. In aquaculture, understanding the interactions existing between the microbiotas of farmed species and their rearing environment could help establish precise bacterial management.

**Method.** In light of these facts, we studied the active microbial communities associated with the eggs and the nauplii of the Pacific blue shrimp (*Litopenaeus stylirostris*) and their rearing water. All samples were collected in September 2018, November 2018 and February 2019. After RNA extractions, two distinct Illumina HiSeq sequencings were performed. Due to different sequencing depths and in order to compare samples, data were normalized using the Count Per Million method.

**Results.** We found a core microbiota made of taxa related to *Aestuariibacter*, *Alteromonas*, *Vibrio*, *SAR11*, *HIMB11*, *AEGEAN 169 marine group* and *Candidatus Endobugula* associated with all the samples indicating that these bacterial communities could be transferred from the water to the animals. We also highlighted specific bacterial taxa in the eggs and the nauplii affiliated to *Pseudomonas*, *Corynebacterium*, *Acinetobacter*, *Labrenzia*, *Rothia*, *Thalassolituus*, *Marinobacter*, *Aureispira*, *Oleiphilus*, *Profundimonas* and *Marinobacterium* genera suggesting a possible prokaryotic vertical transmission from the breeders to their offspring. This study is the first to focus on the active microbiota associated with early developmental stages of a farmed shrimp species and could serve as a basis to comprehend the microbial interactions involved throughout the whole rearing process.

Corresponding authors
Carolane Giraud, cgiraud@ifremer.fr
Nolwenn Callac, ncallac@ifremer.fr

## INTRODUCTION

It is widely recognized that microorganisms, and more particularly prokaryotes, colonize all types of habitats (*Whitman, Coleman & Wiebe, 1998*; *Mora et al., 2011*) and are involved in crucial reactions in various biogeochemical cycles (*Falkowski, Fenchel & Delong, 2008*). Microorganisms have also been extensively studied for their implication in the human microbiota (*Huttenhower et al., 2012*) and thus for their involvement in health, development and behavior (*Cho & Blaser, 2012*; *Moloney et al., 2014*). Marine animals also host essential microbial communities associated with their skin (ex. mammals, fishes) or their exoskeleton (crustaceans) as well as their digestive and respiratory systems (*Apprill, 2017*). These communities are largely influenced by the marine environment which is colonized by all types of microorganisms (*Sehnal et al., 2021*). More and more studies suggest that comprehending the interactions existing among marine animals, their microbiota and their aquatic environment could help understand the response of these animals to climate change and pollution and could also help improve management of farmed species (*Egan & Gardiner, 2016*; *Sehnal et al., 2021*).

In New Caledonia, the Pacific blue shrimp (*Litopenaeus stylirostris*) is a valuable farmed species and represents an economical challenge for the territory (*Beliaeff et al., 2009*). Indeed, since 1970, the sector has become the island's leading food-processing exporter (Rural Agency of New Caledonia) with a semi-intensive annual production of 1,500 tons. Unfortunately, the production has been declining due to seasonal vibriosis touching adult individuals (*Goarant et al., 1999*; *Goarant et al., 2006*) and to larval mortalities that have not yet been explained. Knowing that natural seawater is used during the rearing process and that the New Caledonian lagoon has been more and more exposed to anthropic pressures over the years, hypothesis have been made regarding the water quality in order to explain these early mortalities. Thus, understanding the interactions that exist between the microbial communities associated with the rearing water and the different larval stages of *L. stylirostris* could help lead to precise bacterial management during shrimp farming.

As a first step in understanding the microbiota of *L. stylirostris*, we studied the active microbial communities associated with the eggs, the first larval stages (nauplii) and the rearing water. Samples were collected at three different times in September 2018 (cool season), November 2018 (transitional season) and February 2019 (warm season). Bacterial lineages affiliated to *Aestuariibacter*, *Alteromonas*, *Vibrio*, *SAR11*, *HIMB11*, *AEGEAN 169 marine group* and *Candidatus Endobugula* were found in all the samples but in different proportions suggesting that some bacterial communities in the rearing environment could potentially colonize the eggs and the nauplii. We also found specific taxa related to *Pseudomonas*, *Corynebacterium*, *Acinetobacter*, *Labrenzia*, *Rothia*, *Thalassolituus*, *Marinobacter*, *Aureispira*, *Oleiphilus*, *Profundimonas* and *Marinobacterium* genera in the eggs and the nauplii that were not found in the water samples and could have been vertically transmitted from the breeders.

## MATERIALS & METHODS

### Study design and sample collection

Eggs and nauplii of *L. stylirostris* were supplied by the experimental shrimp hatchery located at the Saint Vincent Bay (Ifremer, Boulouparis, New Caledonia) in September 2018 (M1), November 2018 (M3) and February 2019 (M4). For all experiments, breeders were reared in maturation tanks according to the method described in *Pham et al. (2012)*. Tanks in the maturation and in the hatchery were filled with natural seawater from the Saint Vincent Bay. Natural seawater was pumped through a 1 cm pore size strainer in a primary reservoir (ResI), was then filtered on a 10 μm pore size filter and stored in a secondary reservoir (ResII). In the secondary reservoir, seawater was circulated for 3 days, through a loop composed of 1 and 5 μm pore size filters and a skimmer. Seawater was briefly passed through a UV chamber before filling the hatchery tanks. In each tank, 5 g m$^{-3}$ of ethylenediaminetetraacetic acid (EDTA) were finally added (Fig. S1). Seawater from the primary reservoir was sampled 5 days before artificial inseminations. Seawater from the secondary reservoir was sampled after filtration and skimmer treatments, on the day the eggs were collected. For each experiment, 1L of seawater from the primary and the secondary reservoirs were collected and were filtered on a 0.2 μm pore size filter (S-PAK membrane filter, Millipore). Filters were kept for RNA extractions and were stored at −80 °C; filtrates were used, within the 2 h of sampling, to analyse the Colored Dissolved Organic Matter (CDOM) as described in *Helms et al. (2009)* and *Sadeghi-Nassaj et al. (2018)*).

After artificial insemination and spawning, around a hundred eggs were collected in a 2ml sterile microtube using sterilized pliers. The following day, before transfer into rearing tanks for larval rearing, around a hundred nauplii were sampled in the same way as the eggs using sterilized pliers. All egg and nauplius samples were stored at −80 °C until further RNA extractions. For the M4 experiment, eggs and nauplii were sampled in replicates (M4_Egg1, M4_Egg2, M4_Nii1, M4_Nii2) whereas unique egg and nauplius samples were collected during the M1 and the M3 experiments.

### RNA extractions and sequencing

Due to different sample types, RNA extractions were performed using the RNeasy PowerWater kit (Qiagen) for the filters, and using the RNeasy minikit (Qiagen) for the eggs and the nauplii. Total RNAs were reverse-transcribed into complementary DNA (cDNA) by adding 200 ng of RNAs to a reaction mix (buffer 5X, dNTP 10 mM, random hexamers 50 μm, reverse transcriptase M-MLV (PROMEGA) 200 u μl$^{-1}$, RNAse/DNAse free water). Reverse-transcription was performed in a thermocycler (VerityTM, Applied Biosystems) during 10 min at 25 °C, 2 h at 42 °C and 5 min at 85 °C.

All cDNAs were sent to MrDNA (Shallowater, TX, United States) in order to amplify and sequence the V4 region of the 16S rRNA gene using the 515F/806R primers (*Caporaso et al., 2011*). A HiSeq Illumina sequencing was conducted using a 2 × 300 pb paired-end run.

Due to different sampling times, two distinct sequencings were performed. All samples from the M1 and the M3 (M3a, first sequencing) experiments were sequenced in February

2019 with an average sequencing depth of 20k raw reads per samples. All samples from the M4 experiment as well as the egg and the nauplius samples from the M3 experiment (M3b, second sequencing) were sequenced in November 2019 with an average sequencing depth of 50k raw reads per sample.

## Microbiota analysis

Raw sequences were demultiplexed using the MrDNA tool fastqSplitter (https://www.mrdnalab.com/mrdnafreesoftware/fastq-splitter.html). Demultiplexed sequences were treated using the DADA2 (*Callahan et al., 2016*) package available in the RStudio software (*RStudio Team, 2020*). Briefly, sequences were filtered using a maximum expected error of 2. Filtered sequences were used to estimate error rates and dereplication. Dereplicated samples and error rates were implemented in the DADA2 error model in order to correct sequencing errors and construct Amplicon Sequence Variants (ASVs) as described in *Callahan et al. (2016)* and *Callahan et al. (2016)*. Paired-end reads were merged and chimeras were removed using the consensus method. Taxonomy was finally assigned using the Silva 138 database (*Quast et al., 2013*).

Data were normalized with the Counts Per Million (CPM) method using the cpm function available in the edgeR package under RStudio (*Robinson, McCarthy & Smyth, 2009*). All libraries were normalized to 1,000,000 reads prior to microbiota analysis. A dendrogram based on a Bray-Curtis dissimilarity matrix and Ward method was obtained using the vegan, ggplot2, dplyr and dendextend packages in RStudio (*Galili, 2015*; *Oksanen et al., 2020*; *Wickham, 2016*; *Wickham et al., 2021*). Histogram tables were obtained using the dplyr package in RStudio (*Wickham et al., 2021*). Venn diagrams were made using the open-source component for web environment jvenn (http://jvenn.toulouse.inra.fr/app/example.html) (*Bardou et al., 2014*).

All the 16S rRNA data are available in the NCBI SRA repository (Submission ID SUB9828953, BioProject ID PRJNA736535).

# RESULTS

## CDOM (Colored Dissolved Organic Matter) measurements

Two parameters were analyzed during CDOM measurements of the water samples: the absorption coefficient at 325nm (a$\lambda_{325}$) and the ratio of the spectral slopes (SR) ((a$\lambda_{275-295}$)/(a$\lambda_{350-400}$)) which are respectively used as a proxy for CDOM concentration and as an indicator of the molecular weight of Dissolved Organic Matter (DOM) (*Sadeghi-Nassaj et al., 2018*). The a $\lambda$325 values of all the reservoir samples ranged from 0.8 to 1.1 m$^{-1}$. For all experiments, the SR values varied from 5.98 to 7.67 in the primary reservoirs (ResI) and from 4.79 to 8.12 in the secondary reservoirs (ResII). For the M1 experiment, the SR value was higher in the ResI sample (7.06) than in the ResII sample (4.79). For the M3 and the M4 experiments, trends were inverted (Table 1) as SR values respectively equaled 7.67 and 5.98 in the primary reservoirs; and 8.12 and 6.27 in the secondary reservoirs.

## Microbial diversity

A total of 2,987,759 sequences were obtained from the Illumina sequencings of all samples (M1, M3a, M3b and M4). Sequences were clustered into 4,950 distinct ASVs.

**Table 1  CDOM concentrations and slope ratios in the reservoirs.** Values of CDOM concentration (aλ325) and slope ratios (SR) from 275 to 295 nm and from 350 to 400 nm of the primary and the secondary reservoir (ResI and ResII) samples of the M1, M3 and M4 experiments.

|  |  | ResI | ResII |
|---|---|---|---|
| **M1** | **aλ325 (m⁻¹)** | 0.99 | 1.01 |
|  | **SR** | 7.06 | 4.79 |
| **M3** | **aλ325 (m⁻¹)** | 0.83 | 0.82 |
|  | **SR** | 7.67 | 8.12 |
| **M4** | **aλ325 (m⁻¹)** | 1.06 | 0.98 |
|  | **SR** | 5.98 | 6.27 |

The smallest and largest libraries were respectively composed of 15,399 and 554,056 reads and corresponded to the M3a_Egg and M3b_Nii samples. All libraries were normalized to 1,000,000 reads using the CPM method.

Hierarchical clustering based on Bray–Curtis similarity separated the samples into 2 distinct groups (Fig. 1A). The cluster 1 gathered all the water samples while the cluster 2 contained all the egg and the nauplius samples. All the water samples from the primary reservoirs clustered together with the ResII sample from the M1 experiment. The water samples from the other secondary reservoirs (M3a and M4) clustered separately. Eggs and nauplii sampled during the M3 experiment and sequenced twice formed a unique subgroup while the egg and the nauplius samples from the M1 and M4 experiments clustered together.

For all samples, 11 dominant classes were highlighted (Fig. 1B). Gammaproteobacteria (45% of the total ASV relative abundance) and Alphaproteobacteria (19%) were the most abundant classes followed by Bacteroidia (10%) and Verrucomicrobiae (5%). Acidimicrobiia, Planctomycetes and Cyanobacteriia each represented 3% of the ASVs table. Water samples from all the primary reservoirs showed consistent total bacterial compositions with five predominant classes: Alphaproteobacteria, Gammaproteobacteria, Acidimicrobiia, Cyanobacteriia and Bacteroidia. The bacterial diversity of the M1_ResII sample was very similar to the one of the primary reservoirs. The other water samples from the secondary reservoirs also contained Alphaproteobacteria and Gammaproteobacteria but Planctomycetes dominated the M3_ResII sample while the M4_ResII sample was mainly composed of Verrucomicrobiae. Global bacterial communities associated with the egg and the nauplius samples majorly displayed Gammaproteobacteria, Alphaproteobacteria and Bacteroidia (over 70% of the total ASV relative abundance). Eggs were also composed of Fusobacteriia, especially for the M1 experiment, while nauplii were partly composed of Desulfuromonadia.

## Specific and shared ASVs among samples

In order to highlight specific and shared ASVs among all samples, we constructed several Venn diagrams. First, four distinct Venn diagrams were built to find common ASVs among all experiments for each sample type (Fig. S2). A total of 330 ASVs were shared among all the primary reservoir samples (Fig. S2A). All the secondary reservoir samples owned 215 ASVs (Fig. S2B) whereas the egg (Fig. S2C) and the nauplius (Fig. S2D) samples respectively

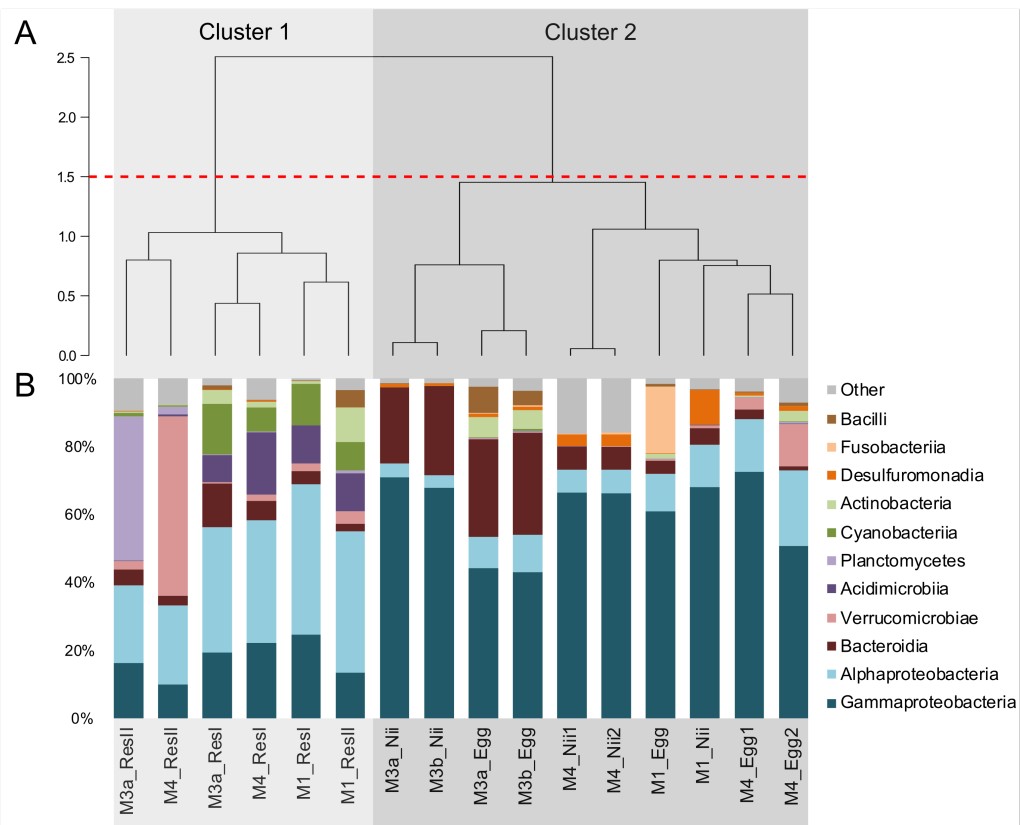

**Figure 1 Bacterial compositions and clustering of all samples.** (A) Hierarchical clustering based on Bray-Curtis dissimilarity. In order to define clusters, a 1.5 threshold was set (represented by the red dotted line). All the water samples are clustered in Cluster 1, in light grey. All the egg and the nauplius (nii) samples are clustered in Cluster 2, in dark grey. (B) Bacterial compositions of the primary reservoir (ResI), the secondary reservoir (ResII), the egg and the nauplius (nii) samples. The highlighted bacterial classes have a total relative abundance higher than 1%.

shared 270 and 259 ASVs. We used these four ASV lists to build a final Venn diagram allowing us to highlight 623 distinct ASVs; specific and common among the compartments (Fig. 2). A total of 257 ASVs were uniquely found in the water samples (Fig. 2A). The primary and the secondary reservoirs respectively showed 145 and 33 specific ASVs while they shared 79 ASVs. All the considered samples shared 62 ASVs (Fig. 2B). In the same way as the water samples, 233 ASVs were only found in the egg and the nauplius samples (Fig. 2C); 88 ASVs were specific to the egg samples while 80 ASVs were uniquely highlighted in the nauplius samples. The eggs and the nauplii shared 65 ASVs that were not identified in the water samples.

## Specific and shared bacterial communities among samples

Based on the Venn diagram (Fig. 2), we analyzed the relative abundance of each subset of ASVs selected above by constructing 100% stacked histograms. The water samples showed specific ASVs (Fig. 3A). The microbiota specifically associated with the primary reservoirs showed 4 dominant groups which accounted for 30% of the total relative abundance
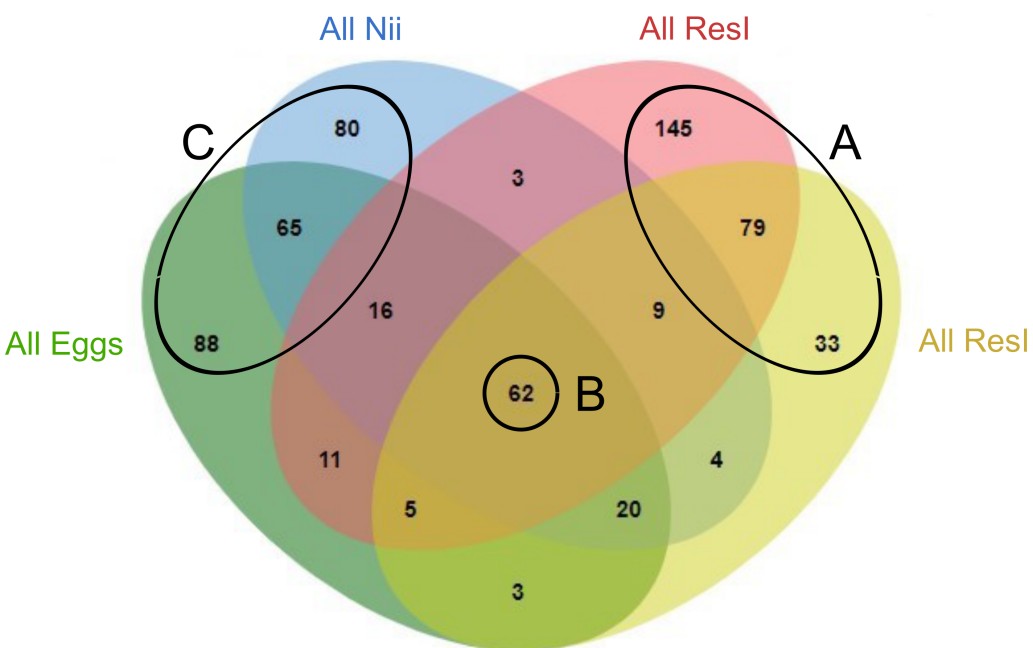

**Figure 2** **Venn diagram of shared ASVs among experiments and sample types.** The green ellipse represents the ASVs common to all the egg samples. The blue ellipse represents the ASVs common to all the nauplius (nii) samples. The red ellipse represents the ASVs common to all the primary reservoir (ResI) samples. The yellow ellipse represents the ASVs common to all the secondary reservoir (ResII) samples. Numbers noted in the overlapping areas correspond to shared ASVs among compartments. Numbers noted outside of the overlapping areas correspond to specific ASVs. (A) Specific and shared ASVs among the water samples. (B) Shared core microbiota among all sample types. (C) Specific and shared ASVs among the egg and the nauplius samples.

(Fig. 3B): *OM60 Clade*, *Sva0996 marine group*, *MB11c04 marine group* and *NS5 marine group*. One ASV was not affiliated to the genus level (*Nitrincolaceae*) and represented 20% of the total relative abundance. *Candidatus Actinomarina*, *Synechococcus* and *SAR116 Clade* were shared between the primary and secondary reservoirs and accounted for 49% of the total relative abundance (Fig. 3C). The secondary reservoir samples highlighted a very specific and highly represented ASV affiliated to *Pedosphaeraceae* family which accounted for 80% of the total abundance (Fig. 3D).

A total of 62 ASVs were shared among all the samples (Fig. 4A). In this microbiota, 15 genera and 4 ASVs which were not affiliated to the genus level were highlighted. Even though all these bacterial lineages were found in all samples, their total relative abundance varied. Indeed, the ResI samples were dominated (60% of the total relative abundance) by *SAR11 Clade Ia*, *HIMB11* and by an ASV affiliated to the *AEGEAN 169 marine group* (Fig. 4B). The secondary reservoirs showed higher abundances for 3 ASVs which were affiliated to *Candidatus Endobugula*, *Gimesiaceae* and *NRL2* (Fig. 4C). In both egg (Fig. 4D) and nauplius (Fig. 4E) samples, *Aestauriibacter*, *Alteromonas* and *Vibrio* genera represented over 50% of the total relative abundance.

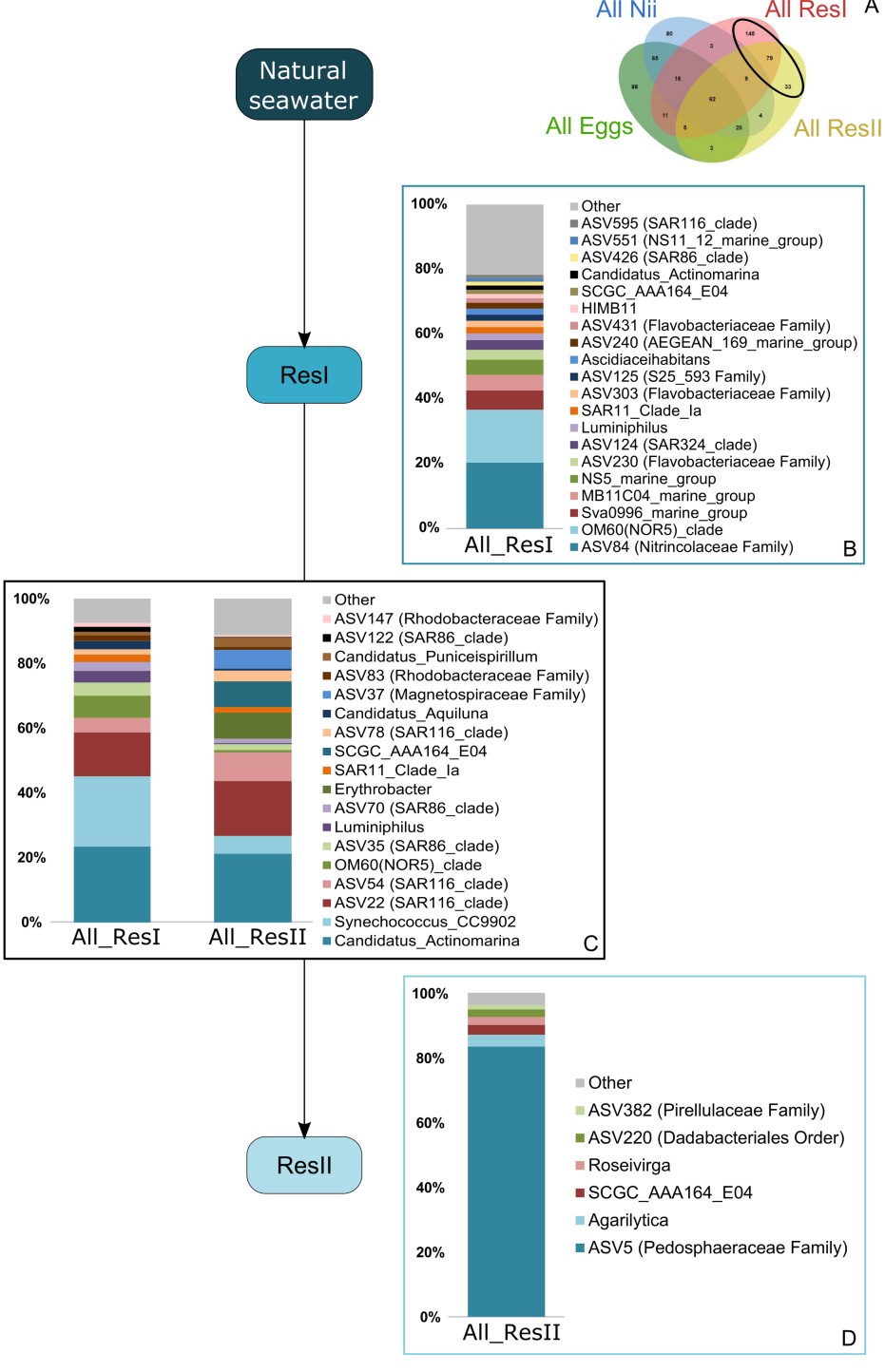

**Figure 3** **Bacterial communities associated with the primary and the secondary reservoirs.** Top bacterial lineages (up to genus when possible) with a relative abundance higher than 1% and specifically found in (A) the primary and the secondary reservoir water samples (ResI and ResII). Histograms were built using the abundance of the selected subset of ASVs. Top bacterial lineages (B) uniquely found in the primary reservoir (ResI) samples (148 ASVs, in red in the Venn diagram), (C) shared between all the water samples (79 ASVs, in the intersection of the red and yellow ellipses) and (D) uniquely found in the secondary reservoir (ResII) samples (33 ASVs, in yellow in the Venn diagram). The arrows represent the potential ASV transmission among compartments.

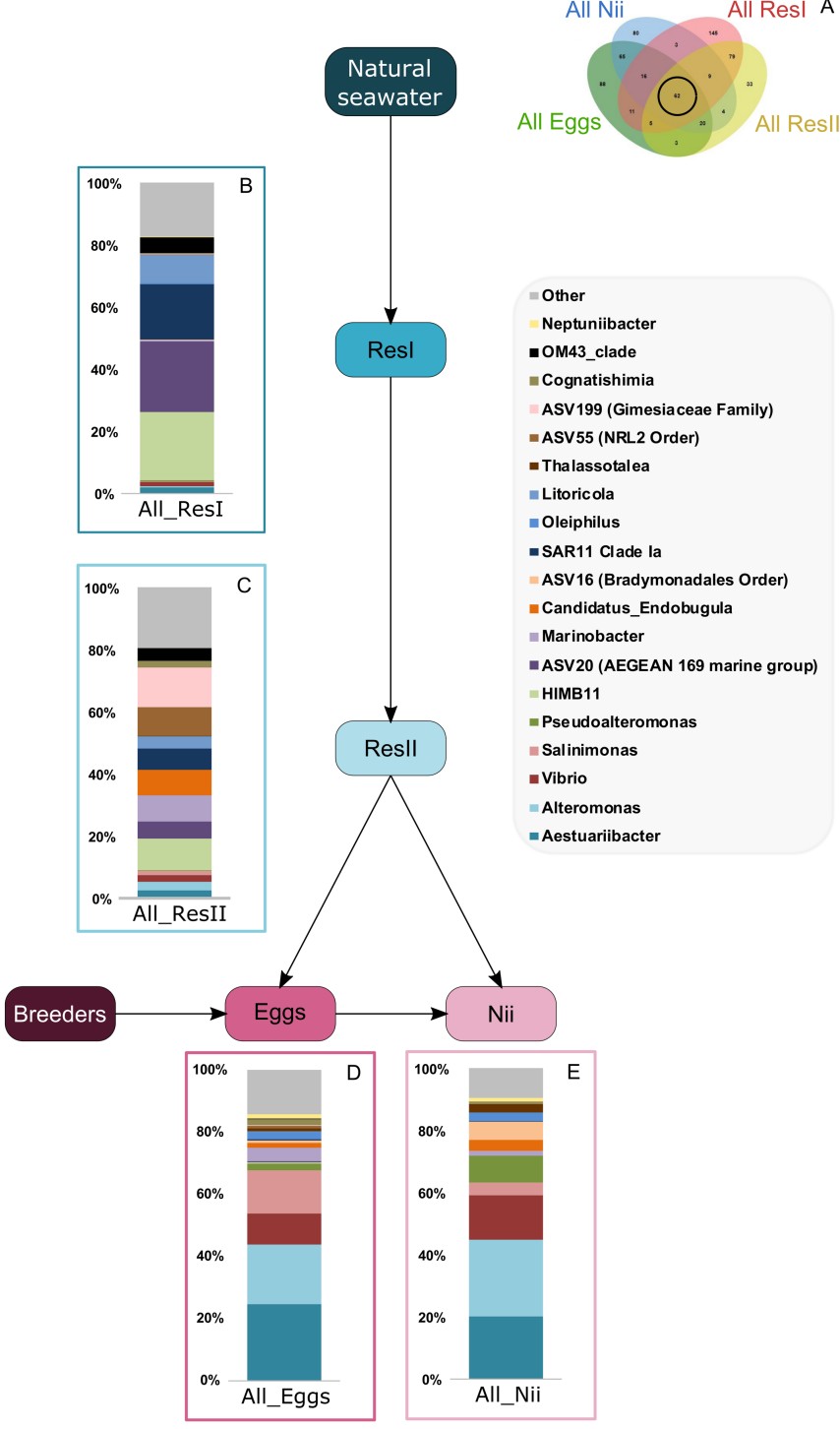

**Figure 4** **Bacterial communities associated with all samples.** Top bacterial lineages (up to genus when possible) composing the (A) shared core microbiota among all samples (62 ASVs). Histograms were built using the abundance of the selected subset of ASVs. The highlighted bacterial lineages have a total relative abundance higher than 1% but are distributed unevenly among (B) the primary reservoir (ResI) samples, (C) the secondary reservoir (ResII) samples, (D) the egg samples and (E) the nauplius (nii) samples. The arrows represent the potential ASV transmission among compartments.

Bacterial compositions of communities specifically associated with the egg and the nauplius samples were evaluated alongside with transmitted ASVs between the two sample types (Fig. 5A). In the egg samples, 21 bacterial lineages accounted for more than 80% of the total relative abundance (Fig. 5B). *Pseudomonas* and *Corynebacterium* each had a relative abundance of 18%. *Acinetobacter* represented 8% of the total relative abundance while *Labrenzia* and *Rothia* each accounted for 6%. These five genera alone represented more than 55% of the total relative abundance among the bacterial communities specifically associated with the egg samples. Some ASVs seemed transmitted from the eggs to the nauplii (Fig. 5C). Among those ASVs, 21 bacterial lineages were highlighted and accounted for 90% of the total relative abundance. The most abundant genera were *Thalassolituus* (20%), *Vibrio* (18%), *Marinobacter* (5%), *Aureispira* (5%) and *Oleiphilus* (4%). One ASV accounted for 5% of the total relative abundance but could not be affiliated to the genus level (*MBAE14* Order). Specific ASVs were also found in all the nauplius samples; 17 bacterial lineages were highlighted and represented 85% of the total relative abundance (Fig. 5D). *Profundimonas* and *Marinobacterium* respectively represented 15% and 8% of the total relative abundance. Two ASVs accounted for 28% of the abundance and were affiliated to SGC AAA286 E23 and Bacteroidia (unknown order).

## DISCUSSION

### Dealing with sequencing depths

Due to different sampling times and sequencing runs, the choice of bioinformatic techniques for data analysis was very important. Thus, we considered and compared several methods to cluster sequences and to normalize the considered dataset. During the last decade, two ways have been used to cluster sequencing reads: the Operational Taxonomic Units (OTUs) or the ASVs. OTUs cluster sequences which are identical up to a fixed similarity threshold (usually 97%) while ASVs regroup identical sequences after a sequencing error correction step (*Callahan et al., 2016*). Comparing the 2 methods, it appears that biological conclusions are very similar (*Allali et al., 2017*; *Glassman & Martiny, 2018*). However, ASVs are advised for improved reproducibility, comparison across studies, meta-analysis and nucleotide-level resolution (*Callahan, McMurdie & Holmes, 2017*; *Porter & Hajibabaei, 2020*). As the samples considered in our study had been collected during 3 different experiments and were sequenced separately, we chose to analyze sequencing data using the ASV method implemented in the DADA2 package under RStudio (*Callahan et al., 2016*) in order to enable comparison among samples. After sequencing data analysis, library sizes among samples varied from 15399 to 554056 reads because of different sequencing depths. As a consequence, hierarchical clustering based on Bray-Curtis dissimilarity of non-normalized data showed a clear sequencing depth effect as samples sequenced together gathered together (Fig. S3). In order to compare samples, data needed to be normalized. The rarefaction method has been extensively used to normalize metabarcoding data but it leads to non-reproducible tables and randomly suppresses OTUs or ASVs (*McMurdie & Holmes, 2014*). The edgeR and DESeq methods have been advised but involve the use of logarithms and do not handle well null values
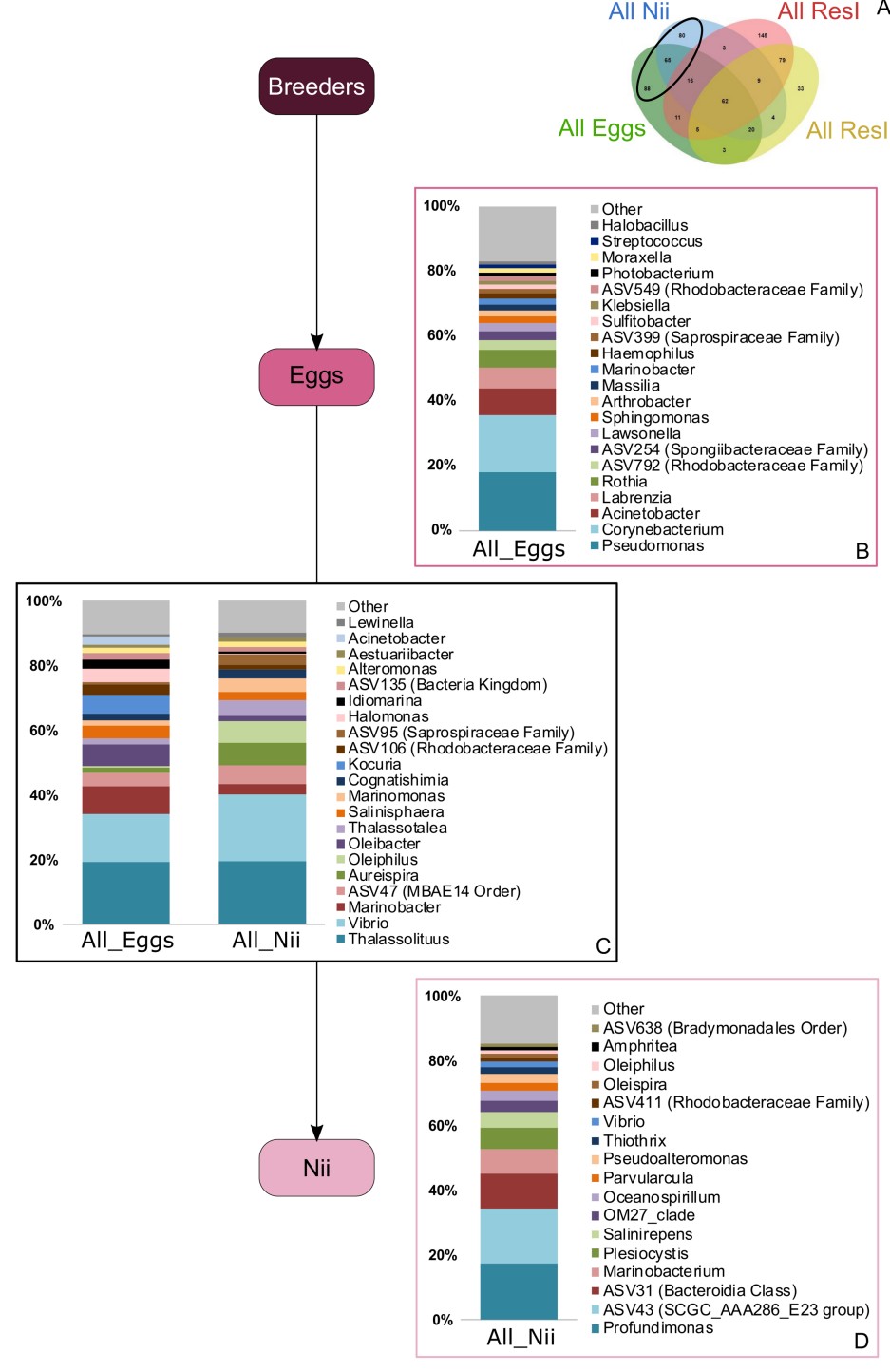

**Figure 5 Bacterial communities associated with the eggs and the nauplii.** Top bacterial lineages (up to genus when possible) with a realtive abundance higher than 1% and specifically found in (A) the egg and the nauplius (nii). Histograms were built using the abundance of the selected subset of ASVs. Top bacterial lineages (B) uniquely found in the egg samples (88 ASVs, in green in the Venn diagram), (C) shared between all the egg and the nauplius (nii) samples (65 ASVs, in the intersection of the green and blue ellipses) and (D) uniquely found in the nauplius (nii) samples (88 ASVs, in blue in the Venn diagram). The arrows represent the potential ASV transmission among compartments.

(*Weiss et al., 2017*) which were quite abundant in our dataset. A recent study showed that simpler methods lead to satisfying normalizations (*Bushel et al., 2020*). Thus, we chose the Count Per Million (CPM) method which enabled hierarchical clustering of the samples according to their bacterial compositions rather than their sequencing runs.

## Chemicophysical and microbial quality of the water samples

CDOM concentrations were constant throughout all the water samples. However, SR values decreased from the primary to the secondary reservoirs during the M1 experiment and opposite tendencies were observed during the M3 and M4 experiments (Table 1). Shifts in SR can be correlated to molecular weight and composition modifications of the dissolved organic matter (*Helms et al., 2009*) suggesting that such changes occurred in the reservoirs and were of different natures during the M1 experiment compared to the M3 and M4 experiments. Overall, all the water samples showed a large abundance of Gammaproteobacteria and Alphaproteobacteria (Fig. 1B) which are respectively dominant classes in pelagic and benthic marine environments (*Zinger et al., 2011*). The primary reservoirs and the M1_ResII sample were also majorly composed of Acidimicrobiia, Cyanobacteriia and Bacteroidia, 3 classes previously identified in pelagic environments and marine ecosystems (*Giovannoni & Stingl, 2005*; *Zinger et al., 2011*). The M3 and M4 secondary reservoirs respectively displayed important abundances of Planctomycetes and Verrucomicrobiae which have been found in soil, freshwater (Australian dam, Hungarian pond) and seawater (the Californian Current) (*Fuerst, 1995*; *Freitas et al., 2012*). CDOM measurements and bacterial compositions in water are known to be correlated factors (*Judd, Crump & Kling, 2006*). A change in the CDOM measurements can result in a change of the microbiota and *vice versa*. Thus, it is quite difficult to distinguish the cause and the consequence. However, both these factors can also be influenced by seasonal variations (*Osterholz et al., 2018*). As our samplings occurred between September 2018 and February 2019, seasonal variations could explain the differences observed in our samples in terms of SR tendencies and differential bacterial communities. However, as the primary reservoirs were quite similar and were a closer reflection of the natural conditions, the differences observed in the secondary reservoirs may also be explained by other factors. As we performed RNA extractions, we studied active prokaryotic communities (*Blazewicz et al., 2013*). We can thus hypothesize that differences in bacterial compositions, and therefore in CDOM profiles, may be due to different active microorganisms throughout the water treatments.

Specific bacterial communities associated with the primary reservoir samples were dominated by 4 marine groups (*OM60 Clade*, *Sva0996*, *MB11c04* and *NS5*) (Fig. 3B). They have been identified as crucial communities in the marine environment and have been found in various ecosystems ranging from marine sediments and surface seawaters to polar and coastal marine ecosystems (*Suzuki et al., 2001*; *Cho et al., 2007*; *Gómez-Pereira et al., 2010*; *Orsi et al., 2016*). Specific ASVs were common to the primary and the secondary reservoirs (Fig. 3C) and were dominated by members of *Candidatus Actinomarina*, *Synechococcus* and *SAR116*. These bacterial groups are ubiquitous in marine ecosystems and are known as or linked to primary producers (*Britschgi & Giovannoni, 1991*;

*Partensky & Vaulot, 1999*; *Ghai et al., 2013*) . This could suggest that these communities are present in the Saint Vincent Bay and persist throughout the reservoirs. However, specific ASVs associated with the secondary reservoirs (Fig. 3D) were largely dominated by an ASV affiliated to the *Pedosphaeraceae* family (Verrucomicrobiae class) suggesting that the water treatments between the primary and secondary reservoirs could select some bacterial communities. This has already been noticed by Vadstein et al. who showed that UV treatment of rearing water induced selection of fast-growing and opportunistic bacteria (*Vadstein et al., 2018*). The water samples considered in this study were collected before addition of ethylenediaminetetraacetic acid (EDTA) in the hatching tanks. To our knowledge, there is very few information about the effect of this metal chelator on active microbial communities in water. Thus, further investigations will be necessary to evaluate the role of EDTA on the microbiotas associated with the rearing water and the animals in shrimp hatcheries.

## Putative factors influencing the microbial communities associated with the eggs and the nauplii of *L. stylirsotris*

In the egg and the nauplius samples, Gammaproteobacteria, Alphaproteobacteria and Bacteroidia dominated (Fig. 1B). These classes have been identified in larvae of the Pacific white shrimp (*Litopenaeus vannamei*) at early developmental stage (*Pangastuti et al., 2009*; *Zheng et al., 2017*) but also in the intestines of adult black tiger shrimps (*Penaeus monodon*) (*Rungrassamee et al., 2013*). Eggs showed members of the Fusobacteriia class which have also been identified in adult shrimps (*Rungrassamee et al., 2016*; *Zeng et al., 2017*) indicating a possible preservation of bacterial communities throughout the whole lifecycle of shrimps. Overall, all the considered samples displayed similar bacterial compositions suggesting that the microbiotas of the eggs and the nauplii could also be influenced by their rearing environments (Fig. 1B). A total of 62 ASVs were shared among all compartments (Fig. 2B) and were considered to be the core microbiota of all our samples (*Shade & Handelsman, 2012*). *Aesuariibacter*, *Alteromonas* and *Vibrio*, which are ubiquitous genera in marine (*Chan et al., 1978*; *Yi, Bae & Chun, 2004*; *Farmer et al., 2015*), were present in all the samples but dominated the eggs and the nauplii (Figs. 4D and 4E). Bacteria affiliated to *Alteromonas* and *Vibrio* produce extracellular hydrolases involved in the biodegradation of various organic carbon sources (*Chan et al., 1978*; *Farmer et al., 2015*). *Vibrios* are also known to express genes involved in the metabolic degradation of chitin (*Hunt et al., 2008*), an important component of the exoskeleton of the nauplii. As mouth opening occurs after the nauplius stage in *Litopenaeus* shrimps (*Wang et al., 2020a*; *Wang et al., 2020b*), the vitellus is the only nutrient and energy resource until the zoae stage (*Harrison, 1997*). Thus, these bacterial communities may be less represented in water and could find favorable organic carbon sources and growth conditions on the external layer of the eggs and the exoskeleton of the nauplii (*Hansen & Olafsen, 1999*). Indeed, the microorganisms that were more represented in the water samples were completely different and belonged to very common marine groups (HIMB11, SAR11, AEGEAN) (*Morris et al., 2002*; *Durham et al., 2014*; *Cram et al., 2015*). As it has been shown for other animal species (brown trout, coho salmon), the total
microbiota of the eggs and the nauplii may embrace the epibiota at the surface but also the endobiota (*Nyholm, 2020*).

Bacterial communities specifically associated with the egg samples (Fig. 5B) were dominated by *Pseudomonas*, *Acinetobacter* and *Labrenzia*, 3 genera found in marine environments (*Fournier & Richet, 2006*; *Palleroni, 2015*; *Raj Sharma et al., 2019*) and which had already been studied during the development of *L. vannamei* and *P. monodon* (*Rungrassamee et al., 2013*; *Wang et al., 2020a*; *Wang et al., 2020b*). The egg samples were also composed of *Corynebacterium*, a genera isolated from coral mucus (*Ben-Dov et al., 2009*). The eggs finally displayed large abundances of *Rothia*, usually found in human mouth cavities (*Tsuzukibashi et al., 2017*) and considered as sample contaminants in vent shrimps (*Methou et al., 2019*). Like *Alteromonas* and *Vibrios*, *Pseudomonas* and *Corynebacterium* are able to degrade different organic carbon sources using extracellular enzymes and could thus find optimal growth conditions with shrimp eggs and the nauplii explaining why they were so abundant among these samples (*Palleroni, 2015*; *Tsuzukibashi et al., 2017*). Concerning *Labrenzia*, they are known to produce labrenzbactin, a catecholate-containing siderophore which demonstrates an antimicrobial activity against *Micrococcus luteus* (*Raj Sharma et al., 2019*). Interestingly, *M. luteus* and *Rothia* are both affiliated to the *Micrococcaceae* family and have been respectively identified in the hepatopancreas of healthy adult shrimps (*L. vannamei*) and in the eggs of vent shrimps (*Durán-Avelar et al., 2018*; *Methou et al., 2019*). This suggests that *Rothia*, may not be a sample contaminant as previously stated by *Methou et al. (2019)*. Further investigations will be necessary in order to determine if this genus is part of the core microbiota of the eggs of *L. stylirostris* or not. Either way, all the ASVs that were not identified in the water environment could have been potentially vertically transmitted from the breeders. Vertical transmission has also been suggested in hydrothermal shrimps where the eggs were associated with specific bacterial communities that were not found in the environment (*Methou et al., 2019*). Intraovum vertical transmission has already been highlighted in farmed fish eggs, accounting for 20% of the total bacterial relative abundance in eggs (*Hansen & Olafsen, 1999*). Furthermore, a vertical transmission of 2 virus types has been proved using molecular techniques in the redclaw crayfish cultured in Australia (*Jaroenram et al., 2021*). Thus, supporting our hypothesis of a potential vertical transmission in the Pacific blue shrimp.

Just as the eggs, the nauplii also displayed specific bacterial communities that were not found in the other samples (Fig. 5D). The most abundant genera were *Profundimonas* and *Marinobacterium* which are both aerobic Gammaproteobacteria isolated from seawater samples (*González et al., 1997*; *Cao et al., 2014*). *Marinobacterium* have also been highlighted in pipefish as a potential beneficial bacterium to larvae (*Beemelmanns et al., 2019*) and some strains are also known to hydrolyse chitine suggesting that they could play an important role in growth at nauplius stage. These ASVs were specifically identified in the nauplius samples and could have been vertically transmitted as well as for the eggs. Once again, as we studied active microbial communities, we can hypothesize that some bacterial communities are acquired from breeders and are activated at different developmental stages.

The nauplius samples could also be under the influence of bacterial communities associated with the eggs as specific ASVs seemed to be transmitted between the 2 sample types (Fig. 5C). These communities were mainly composed of *Vibrio* and *Oleiphilus* which have been identified in several marine environments and animals (*Golyshin et al., 2002*; *Farmer et al., 2015*). *Aureispira* bacteria were also abundant. They have been isolated from marine sponges and algae and are thus frequently found in marine organisms (*Hosoya et al., 2006*). *Marinobacter* and *Thalassolituus* have been isolated in farmed shrimp experiments (*Dineshkumar et al., 2014*). *Thalassolituus* has also been identified at the nauplius stage of *L. vannamei* and has been suggested as a beneficial bacterium for larval development (*Wang et al., 2020a*; *Wang et al., 2020b*). *Vibrio* and *Thalassolituus* have also been identified in healthy adult shrimps (*Rungrassamee et al., 2016*; *Wang et al., 2020a*; *Wang et al., 2020b*) supporting our previous theory that some bacterial communities may be acquired from the breeders and kept throughout the whole lifecycle. Even though *Vibrio* sp. can be associated with diseases, they have been identified in adults and nauplii of *L. vannamei* shrimps regardless of their health status (*Vandenberghe et al., 1999*; *Cornejo-Granados et al., 2017*). Some species are even known to be selected in the environment by their host as symbionts (*Nyholm & McFall-Ngai, 2004*).

## Possible dynamic interactions among compartments

Interestingly, five genera (*Vibrio*, *Marinobacter*, *Thalassotalea*, *Alteromonas* and *Aestuaribacter*) were found in the ASVs shared between the eggs and the nauplii (Fig. 5C) but also in the ASVs shared between the water reservoirs and the animals (Fig. 4). *Marinobacter* and *Aestuariibacter* have been identified in the rearing environment of adult *L. vannamei* and brackishwater shrimps (*Chen et al., 2019*; *Dineshkumar et al., 2014*). *Vibrio*, *Alteromonas* and *Thalassotalea* have all been spotted in *L. vannamei* at different developmental stages as well as in the surrounding water (*Wang et al., 2020a*; *Wang et al., 2020b*; *Zhang et al., 2014*; *Zheng et al., 2017*). In our study, these genera were highlighted from different ASVs in different compartments, suggesting that some taxa may be acquired from the aquatic environment by the eggs and the nauplii but also by the breeders which then potentially transmit them to their offspring, supporting the hypothesis of a possible vertical transmission of bacterial communities in *L. stylirostris* shrimp larvae (Fig. 6). This also shows that the microbiota associated to aquatic organisms and to their eggs are shaped by complex interactions between environmental and transmitted microorganisms (*Sylvain & Derome, 2017*).

## CONCLUSIONS

To our knowledge, this study is the first to focus on the active microbiota of the eggs and the nauplii of a farmed shrimp species while considering the impact of the rearing water. Taken together, our results provide evidence of a core microbiota (*Aestuariibacter*, *Alteromonas*, *Vibrio*, *SAR11*, *HIMB11*, *AEGEAN 169 marine group*, *Candidatus Endobugula*) among all samples suggesting a microbial transmission from the surrounding environment to the animals. As several ASVs were co-owned between the eggs and the nauplii, we also highlight a possible vertical transmission between the two compartments and potentially from the

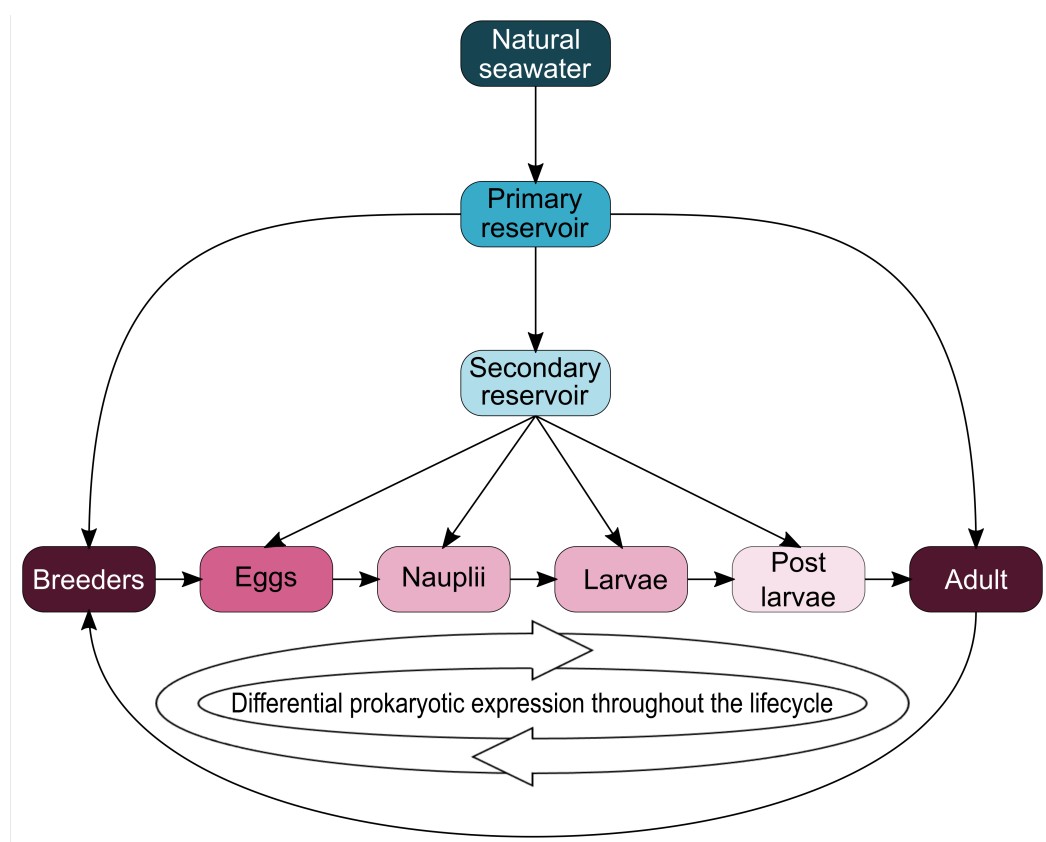

**Figure 6** **Schematic representation of potential microbial transmission and expression throughout the lifecycle of Litopenaeus stylirostris.** The arrows represent potential lineage transmissions.

breeders as specific ASVs were not found in the rearing water (affiliated to *Pseudomonas, Corynebacterium, Acinetobacter, Labrenzia, Rothia, Thalassolituus, Marinobacter, Aureispira, Oleiphilus, Profundimonas, Marinobacterium*) (Fig. 6). This last point has not yet been confirmed as several male breeders are used to inseminate several females which produce eggs that are pooled before larval rearing, thus making it quite difficult to determine the genealogy of the shrimps and their offspring.

## ACKNOWLEDGEMENTS

We are grateful to the SASV teams for their help throughout all the experiments: zootechny and laboratory support. We especially thank Jean-Sébastien Lam and Julien Le Rohellec for their precious help in the hatchery; Florence Antypas and Etienne Lopez for their contribution; as well as Valentine Ballan and Gwenola Plougoulen for their valuable biomolecular work.

### Funding

This work was supported by the RESSAC project (LEAD-NC, Ifremer New-Caledonia) within the framework agreement with the New Caledonian Provinces and Government and by the Pacific Doctoral School. The funders had no role in study design, data collection and analysis, decision to publish, or preparation of the manuscript.

### Grant Disclosures

The following grant information was disclosed by the authors:
The RESSAC project (LEAD-NC, Ifremer New-Caledonia) within the framework agreement with the New Caledonian Provinces and Government and by the Pacific Doctoral School.

### Competing Interests

The authors declare there are no competing interests.

### Author Contributions

- Carolane Giraud analyzed the data, prepared figures and/or tables, authored or reviewed drafts of the paper, and approved the final draft.
- Nolwenn Callac conceived and designed the experiments, performed the experiments, analyzed the data, prepared figures and/or tables, authored or reviewed drafts of the paper, and approved the final draft.
- Maxime Beauvais performed the experiments, analyzed the data, authored or reviewed drafts of the paper, and approved the final draft.
- Jean-René Mailliez, Dominique Ansquer and Dominique Pham performed the experiments, authored or reviewed drafts of the paper, and approved the final draft.
- Nazha Selmaoui-Folcher conceived and designed the experiments, authored or reviewed drafts of the paper, and approved the final draft.
- Nelly Wabete and Viviane Boulo conceived and designed the experiments, performed the experiments, authored or reviewed drafts of the paper, and approved the final draft.

### Data Availability

Data is available at NCBI SRA repository under the submission ID SUB9823633 and BioProject ID PRJNA736535.

### Supplemental Information

Supplemental information for this article can be found online at http://dx.doi.org/10.7717/peerj.12241#supplemental-information.

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
