# Peer review of "Potential lineage transmission within the active microbiota of the eggs and the nauplii of the shrimp Litopenaeus stylirostris: possible influence of the rearing water and more"

_PeerJ, doi:10.7717/peerj.12241_

## Round 0.1 · original submission · Minor Revisions

Please provide your revised manuscript along with a point-by-point reply to all of the reviewers comments.

Reviewer 1 ·

Basic reporting

The study by Giraud and colleagues focuses on the active fraction of eggs and nauplii microbiomes of Litopenaeus stylitostris, also with the aim of understanding the role of the environment in conferring microorganisms, as well as the potential vertical and horizontal transmission of microbial taxa. The manuscript is overall novel, well written, in professional and unambiguous language, and the scientific aims and conclusions are well reported and discussed; the topic of this study is certainly of interest for the readership of this Journal. I only suggest some minor comments, described in more detail below.

Experimental design

I suggest the Authors to add some clarifications on the study design; for instance, it would be helpful to have a scheme (also as Supplementary Material) of the tanks system (ResI, ResII, filtration sizes etc). At lines 137-139 I suggest to add references for the packages used to run analyses.
Line 159: It would be useful to have an idea of the range of ASVs found in the analyzed samples, for example by adding min-max ASVs values at the end of this sentence.
Line 175: Plancomycetes
Lines 234-248: In my opinion, this paragraph it’s excessively long and could be moved in the M&M/results sections.

Validity of the findings

Results and conclusions are well supported and discussed. I only suggest to remove L403-405 from the Conclusions paragraph.

Reviewer 2 ·

Basic reporting

1. The structure of the manuscript follows the journal standards.

2. There are some grammatical, linguistic and spelling mistakes that should be revised.

3. The authors are recommended to re-check Peer J authors instructions about the reference guidelines

4. I could not access the raw data, please check the accession number. No results were found in NCBI SRA repository for the following submissions “Submission ID SUB9828953, BioProject ID PRJNA736535”

Experimental design

There are few major issues with the experimental design that prevents from completely supporting the claims and conclusions made in the manuscript.

Please provide more information’s/details about the samplings (time points, sample size, numbers of replicates, water sample volumes filtrated) in all experiments (M1, M3, M4). In the current version is not clear which type of samples have been collected in each experiment. Please also describe the way the nauplii have been collected.

Furthermore, please justify the use of 2 extraction kits.

Please explain why the Eggs and nauplii sequenced twice (Lines 164 – 165)

I advice the authors to give some information’s to the readers about the ASVs clustering at the materials and methods section.

Validity of the findings

1. The normalisation method it is not needed to be mentioned in the conclusion section.

2. There are also recent studies suggesting vertical transmission in farmed fish eggs and are worth mentioning

3. The results are relevant results to hypotheses.

·

Basic reporting

The manuscript is globally well written although I have a few comments:
- from line 344 to 353: this section lacks structure and some sentences are hard to understand (e.g. line 344). I would suggest to rewrite this section and avoid starting sentences with "As for...". Also, in line 352, a reference is needed for the characterization of Rothia in vent shrimp eggs.
-line 364: spelling mistake "beneficial bacteria" not "benefic bacteria"
Also it's either "potential beneficial bacteria" or "a potential beneficial bacterium".
-line 141. The authors mentioned a submission ID and BioProject ID. However, no raw data is available under these codes. Please make sure the raw data have been released in the NCBI SRA repository.

Experimental design

Replicates have only been used for the last experiment (M4). In addition, there are no samples from the water in tanks when collecting eggs and nauplii samples. It is likely that the bacterial composition of the seawater in tanks is different from the reservoirs, in particular following the addition of EDTA. As such, there could be bacteria in the seawater of tanks that were not detected in the seawater of the reservoirs, potentially overestimating the number of specific ASVs found uniquely in the eggs and nauplii. This is an important bias for the generation of the Venn diagrams which most results in this study are based on. While this issue is mentioned in l302, it could have been avoided by collecting seawater samples during the collection of eggs and nauplii.

Validity of the findings

As it is currently described in the legends and in the main manuscript, it sounds like the bar graphs (in Figures 3, 4, and 5) represent the relative abundance of specific ASVs in relation to all ASVs. However, they actually represent the relative abundance of a subset of ASVs (those selected from the Venn diagram). The actual relative abundance of these ASVs is far less than what is displayed in the bar graphs since many ASVs are excluded for the generation of these graphs. This needs to be better explained in the manuscript and the figure legends.

---

## Round 0.2 · accepted · Accept

Thank you for addressing all of the reviewers' comments.